# OpenReview forum: "Reproducibility Study of "Cooperate or Collapse: Emergence of Sustainable Cooperation in a Society of LLM Agents""
_TMLR — Rejected by TMLR_

### Review · Reviewer_d2Te · 2025-06-13

**Summary Of Contributions:**

The paper is a reproducibility study of "Cooperate or Collapse: Emergence of Sustainable Cooperation in a Society of LLM Agent" (Piatti et al., '24), a work that introduces GovSim, a simulation framework that evaluates the behavior of LLM agents in resource-sharing problems. In summary:
* The paper validates the core claims of Piatti et al., '24 in one of three scenarios (Fishery) that cooperative behavior emerges in more capable models, and that "universalization" via instructing the model to consider if all agents behave identically can also improve cooperation.

The paper further makes four extensions:
* Evaluation of other models, with similar results.
* Evaluation on Japanese-language instructions, which is based on the assumption of "Japanese cultural emphasis on group harmony" (Section 3.3.2) with similar results to the default English-language scenarios.
* The paper extends evaluation to a loss-aversion scenario, flipping the resource-sharing task to minimize (rather than maximize) the quantity of a resource.
* The paper extends evaluation to scenarios with agents of differing capabilities (i.e., differing weak + strong models)

**Audience:**

Yes

**Broader Impact Concerns:**

No additional concerns.

**Claims And Evidence:**

No

**Requested Changes:**

The following are critical for a recommendation for acceptance.
* On the extension with additional models:
    * Provide specific hypotheses/justification for any additional models tested beyond the original paper. I.e., for open-weight models, what are some of the architectural/modeling differences in these models, and why might we expect them to have/not have an effect on the results?
* On the cross-lingual extension:
    * Provide additional evidence for cross-linguistic behavioral consistency by adding analogous experiments on more language(s), perhaps prioritizing those believed to be more prevalent in training data.
    * Further discuss any validation steps taken to ensure the quality of the translated prompts, expanding on Appendix G.2. It is similarly unclear what it means to keep the prompts "culturally neutral," so clarification would also be appreciated as well. If gold-standard evaluation is not possible, report results for an automated approach (e.g., back-translation + some scoring function such as BLEU or LLM-as-judge [1]).

I might even find it more convincing to drop those extensions altogether (additional models + cross-lingual evaluation), in favor of deeper experiments analyzing the inverse (trash) scenario + agents with differing capabilities cooperating.

The following would strengthen the paper but are not critical to securing my recommendation:
* Revising Table 1 to also include a comparison to the original results, to verify that the replication is indeed close to the original.
* Revising Table 3 to be relative to the default scenario (and the current Table 3 could go in the APpendix, perhaps), to make the claim "while most models failed the sustainability test in the default setting, nearly all succeeded in the trash scenario" more clear.
* In the expeirment combining higher + lower performing models, it would be interesting to try and detect interactions where the weaker model is instructed by the higher-performing model to be aware of its overconsumption, and to test if these interactions are predictive of the stronger model "steering" the weaker model to behave more sustainably. The claim is worded a bit informally as-is and would benefit from more quantitative evaluation.
* There's some misformatted bibliography entries where et al is used — entries should read "Lastname, Firstname et al." instead of "Lastname et al., Firstname."

[1] Edunov et al. On The Evaluation of Machine Translation Systems Trained With Back-Translation. ACL 2020.

**Strengths And Weaknesses:**

# Strengths
* The papers' replication of core claims is sufficiently thorough.
* I found the motivation for the extension to a loss-aversion scenario and agents with differing capabilities to be highly interesting and well-motivated.
    * It's interesting that flipping from resource maximization to minimization qualitatively changed model behavior — I particularly like this finding: "While most models failed the sustainability test in the default setting, nearly all succeeded in the trash scenario."
* I like the attempts the engage with literature outside of CS in social psychology to motivate and ground experiments, framing the work as tests of whether LLMs exhibit "inter-group" behaviors previously studied in human-human interaction.
* The paper is well-written and easy to follow.

# Weaknesses
* The motivation for the first extension to evaluation of other models is unclear. While I credit the work for evaluating an impressively large quantity of models, I would rather see fewer models/model families w/ clearer hypotheses/justification for inclusion of the models. One hypotheses the authors could explore, for example, is: are there other axes, besides "capability," that could yield models that engage in more resource-sharing than non-reasoning models (e.g., "reasoning" models)?
    * I think this is already present to some extent with the inclusion of Deepseek V3, but the addition of other model families such as Qwen-2.5 is not as well-motivated.
* Model "capability" is not well-defined. Capability can be measured in multiple ways, according to multiple benchmarks — is there a particular sense in which "capability" is being used in the paper?
* The motivation for the Japanese-language experiment is also unclear, and there are some missing details.
    * First, to support the premise that translation into Japanese could promote collectivist behavior in LLMs, it's quite strange to cite a work that seems to caution specifically against cultural stereotyping, especially a book that uses perceptions of Japanese collectivism as a case study. Premise notwithstanding, from the technical side, it's unclear whether the lack of difference between the English vs. Japanese experiments are due to (1) the hypothesis of cultural collectivism being incorrect altogether, or (2) the hypothesis of cultural collectivism being correct, but not necessarily a prominent/detectable linguistic feature, or (3) the hypothesis of cultural collectivism being correct and detectable, but the models are trained with insufficient Japanese-language data to "internalize" such values in their behavior.
    * Second, how the instructions were validated is also unclear, as translation is not always an objective task: variations in style/formality/tone in one language (e.g., Japanese) may all map to the same instruction in English, for example, yet affect the behavior of the LLM meaningfully. As as a minimum, one should report example LLM interactions and prompts in *both* English and Japanese for transparency.
    * Lastly, experiments on one language are insufficient to conclude that "Cooperative behavior exhibits remarkable stability across linguistic boundaries" (Section 5.2).

---

### Review · Reviewer_We6J · 2025-06-16

**Summary Of Contributions:**

This paper reproduces and extends a handful of previously published results on cooperation in multi-LLM agent scenarios.
In particular, they use the GovSim framework to look at shared resource-management scenarios such as coordinating fishing of shared lake which risks being overfished and depleted.
In the reproduction segment of the paper, their results align with the original work showing that agent groups based on more capable LLMs (high parameter count, tending to be closed-weight) tend to do better than smaller, open-weight models (i.e., not deplete the shared resource).
The extensions to prior work include investigating a negative framing of the same shared-resource problem as well as mixing more and less capable models in this same experiment.
These results show, first, that negative framing leads to an increase in performance in some low-performing models, and second, lower-performing models can be influenced by higher-performing ones to cooperate and use the shared resource more efficiently.

**Audience:**

Yes

**Broader Impact Concerns:**

I think the main broader impact concern is risk associated with mixing more and
less capable agents where the stronger agents could influence the weaker ones
to the latter's detriments.  This was addressed adequately in Sec F.1 in the
appendix.

**Claims And Evidence:**

Yes

**Requested Changes:**

- [Sec 3.3] The claim that "[t]his theoretical foundation ensures that our findings contribute to broader understanding of cooperative behavior rather than simply documenting empirical observations" is over-stated so far as I can see.  While the references to loss aversion and social influence are helpful and relevant, the level of detail that the study goes into does not establish these effects as being the cause of behavior.  The results certainly align with the hypothesis, but there are a number of confounding factors (e.g., prompts are not minimally different) that, I believe, require this claim to be attenuated.
- [Sec 3.3.3] Is there a description of this somewhere in the paper beyond "this is just isomorphic to the fishery environment"?  This alone does not give me a clear idea of how the environment works.
- How do you define higher- and lower-performing models?  Is this just in terms of how they do on the baseline fishery scenario or does it refer to some other characteristic (e.g., benchmark performance, parameter count)?

### Minor
- [Sec 1.] "Tragedy of the Commons": fix the quotes (i.e., use backticks to open quotes and apostrophes to close them) (problem shows up with most quoted phrases).
- [Table 1] Add a `\midrule` above "Closed-Weights Models" since it is a bit difficult to read as is.

**Strengths And Weaknesses:**

### Strengths
- The variations presented on the baseline fishery environment are interesting and relevant to better understanding and using LLMs in multi-agent scenarios.
- The results are promising insofar as they both show improved performance without the need for increased complexity or resource utilization.
- Aside from the lack of clarity in the negative-framing environment (mentioned below), the methods and results are clearly explained and easy to follow.
### Weaknesses
- In the negative framing scenario, it is not clear if the change in behavior was due to the polarity of the framing other factors since multiple aspects of the scenario change.
- While the theoretical "grounding" for the extended experiments are useful to mention, I do not think the experiments are able to establish that phenomena are indeed present (or absent). A more thorough analysis of both the theoretical foundations and the experiments themselves would be required for such claims to be solid.
- It is not clear how the negative-framing environment works (mentioned further in _Requested Changes_).

---

### Review · Reviewer_hQQx · 2025-07-10

**Summary Of Contributions:**

This paper is a replication study of a paper studying the behavior of LLMs in a resource-sharing setting, where a group of LLMs must consume resources from a shared pool. The LLMs participate in successive iterations of consumption and discussion, with the aim of cooperating over time.

First, the authors replicate a subset of results showing that smaller models fail to sustain resources, but more complex models can sustain resources, focusing on a single scenario of the three from the original paper. Then, the paper extends the results along several dimensions, including varying the language of the model (to explore the potential of embedded cultural values effects), changing the task from a positive to a negative framing (to explore possible loss aversion effects), including LLMs with qualitatively different architectures, and using groups of LLMs with different capabilities. The paper explores these questions empirically, providing prompts to the LLMs and then observing and comparing the output of successive rounds of resource-sharing between agents in these different settings. This work finds that:
- the original results relating to the size generalize,
- the results generalize to qualitatively different LLM architectures,
- varying the language doesn't have a large impact,
- a loss framing makes agents more cooperative, and
- a group heterogeneous agents with enough skilled agents can sustain cooperation.

**Audience:**

Yes

**Claims And Evidence:**

No

**Requested Changes:**

First, it would be very helpful if the authors could address my questions about the loss aversion extension. Ideally, the authors could have results in a scenario that holds constant all aspects of the original resource-sharing setting except for sharing the framing from gains to losses (or argue from related literature or first principles that the trash example doesn't have the issues I'm concerned about.) It would also be helpful if the authors could provide the prompt for the trash scenario. I've responded "No" to the "Claims and Evidence" prompt mainly for this reason; I'm not convinced from the paper that the "loss aversion" experiment actually shows effects of loss aversion.

I would also recommend the authors to provide more clear and detailed descriptions of their methodology wherever possible.

It would also be very helpful to validate these extensions in another setting. However, LLM empirics is not my research focus, so I am not sure what the standard of evidence is / how reasonable this request is, and defer to more experienced reviewers.

**Strengths And Weaknesses:**

### Strengths

The original paper asks whether LLMs can collaborate in resource-sharing contexts: the extensions examined enrich and contextualize the answer to this question along several natural and important dimensions -- that is, if one intends to deploy LLMs in resource-sharing contexts, these are key axes along which one would wish to understand the LLM's behavior. I especially appreciate the result examining the outcomes with heterogeneously-sized models-- the implication of the result that the authors give (that carefully developing heterogeneous teams of LLMs for resource-sharing could perform well while saving computational costs) is a very interesting suggestion.

### Weaknesses

The methodology in Section 3 for the extensions lacks sufficient detail. In particular:
- What is the prompt used for the loss framing task and the Japanese-language task? For the Japanese-language extensions, which models in particular were selected? What was the human review process for the prompt?
- For the heterogeneous agent description (3.3.8), the "Behavioral Analysis" and "Control Comparisons" sections are very vague.

While the trash scenario changes the resource sharing problem from maximizing gains to minimizing losses, it also changes the type of task (although, because the prompt is not provided, I don't know exactly what the trash elimination task involves). Why can we conclude that the change in behavior is due to the "loss framing" rather than the "trash framing" (trash removal, as noted by the authors, is a common problem which households solve using rotating schedules, which is the solution found by the agents)? If the example had been some other loss-related resource sharing problem where a rotation-based solution isn't common in real settings, would the agents still find a rotation-based solution? Would they still collaborate?

This weakness is exacerbated by the fact that the authors only examine a single scenario (as opposed to the original paper which examines three scenarios) -- in this case, it's difficult to tell the extent to which the extensions are an artifact of the particular setting/prompt. This is true for all of the extensions, though it is the most salient for the loss aversion setting.

Minor:
- The order the extensions are presented and discussed varies through the paper. It would be helpful for readability if the extensions were discussed in the same order in each section.

---

> ### Author Response · Authors · 2025-07-10
>
> We would like to thank the reviewer for taking the time to read our paper carefully and for the thoughtful suggestions, which have helped us improve the paper.
>
> --
> - Regarding the Japanese extension, following Reviewer 1’s advice, we decided to remove it from the paper. A proper analysis would require more extensive experimentation, involving multiple models, languages, well-defined prompts and translations, as well as additional ablation studies. Due to limited human and computational resources, this was not feasible. Instead, we chose to concentrate on the inverse and heterogeneous scenarios, aiming to contribute meaningful insights within our current capabilities, i.e., __do more with less__.
> - As a result, we have expanded the analysis and provided more context for both the inverse and heterogeneous environments (see Sections 3 and 4). We now include the full prompt used for the inverse scenario in Appendix C, and a sample dialogue from that environment is shown in Section 3, providing full evidence of the results.
>
> --
>
> - While we agree that additional inverse scenarios would further strengthen the findings, we had to leave those for future work due to resource/computational constraints, which are not feasible in our research setting. Nevertheless, we want to emphasize that the shift from task framing to loss framing involved more than a superficial change. Specifically, the task structure only changes terms such as “fish” to “trash” and “village” to “household.” In contrast, the loss framing introduces meaningful structural differences, such as explicitly stating that “each unit of trash taken out costs time and resources” and highlighting long-term sustainability goals. These differences are outlined in Figure 7, the prompt.
> - We believe the appearance of rotation-based strategies is more strongly linked to the loss framing than the specific theme of trash removal. In real-world households, rotation tends to be a pre-arranged or externally imposed solution. In our simulation, however, rotation emerged gradually and only under the threat of future collapse, through agent negotiation across rounds. It was not always observed, but when it did occur, it reflected the need for agents to take turns sacrificing their own short-term utility for the long-term stability of the group. We believe this cooperative behavior would likely emerge in other loss-framed scenarios as well, even though the solution might not be a “rotation”.  This interpretation is also supported by the fact that task-specific terms (such as “trash” or “household”) make up only about 1% of the prompt. And, as shown by Piatti et al., model performance remains stable across different task framings, suggesting that behavior is shaped more by the underlying goal and positive nature of the resource than by superficial content like resource name or location setting. Beyond this aspect, even when rotation does not emerge, agents in the loss-framed setting often volunteer to take on a larger burden to prevent collapse. In contrast, in the gain-framed fishery scenario, agents (especially smaller models) are less willing to compromise their own gains, quite often resulting in collapse (for example, GPT-4o-mini, which always collapses in the fishery and not on trash). This highlights a key asymmetry: models appear to weigh the cost of losses more heavily than the benefit of gains.
>
> --
>
> - A more detailed discussion of the previous point has been included in Section 4.
> - We revised the ordering of the extensions to ensure a consistent thread.
> - We added further detail to the theoretical framework and methodology across Sections 3 and 4 to clarify our experimental setup for both the inverse and heterogeneous environments.

---

> > ### Comment · Reviewer_hQQx · 2025-07-21
> >
> > Thank you for your response! I have a few thoughts, and some follow-up questions.
> >
> > ### Trash scenario prompt
> >
> > Taking a look at Figure 7 (the loss framing prompt), I feel that the differences are quite significant.
> > 1. First, the loss framing changes "For every ton of fish caught, a fisherman earns one thousand dollars" to "Each unit of trash taken out costs you time and resources". This changes the framing from a well-defined, linear utility function, to an underspecified, (potentially non-linear) utility function. This seems like it could cause differences beyond the gain vs. loss framing.
> > 2. The number of fish in the fishery scenario doubled each month, but the trash increases by a fixed amount, so the dynamics of the problem change between the settings as well.
> > 3. Why do you highlight the sustainability implications more in the trash example?
> > 4. There is also a very minor change in modifying "the beginning of each month" to "the start of each month"; was this necessary, and if so, why?
> > (I'm referring to Figure C.1 of Piatti et al. for the original Fishery Scenario prompt, please let me know if I should refer to a figure in your paper instead, if you changed the prompt for the baseline fishery scenario at all.)
> >
> > ### Loss vs. task framing
> >
> > I don't agree that because the task framing only takes a small portion of the text, it must have a small impact on the results. When reading a piece of text, words that signal the topic/context (such as the task) are highly important for understanding the text, and I would naturally expect a language model to reflect this. Moreover, the loss framing aspect also only affects a small portion of the text (I believe this is a relatively _lower_ portion of the text than the task framing; 19/225 words in the prompt are from {trash, house{hold}, roommate(s)}, which is over 8%.)
> >
> > ### Additional questions
> >
> > During the dialogue period, do the models all respond at the same time, or respond in sequence? Why does it appear in Figure 9 that the agents aren't "listening" to each other in the conversation, but are repeating each others' ideas (i.e, why do Jack and Luke both suggest rotation without acknowledging that Emma also just suggested this)?

---

> > > ### Author Response · Authors · 2025-07-27
> > >
> > > - The move from a linear gain (“$1000 per ton of fish”) to a cost-based loss framing (“time and resources”) indeed removes an explicit numeric incentive. This was intentional, as we wanted to avoid prescribing a fixed utility-maximization strategy and instead allow agents to reason about burden and sustainability. Additionally, having such values can influence the model behavior, spending 10min might be disregarded while spending 1000min can have a huge impact, which is hard to evaluate and requires a full study on that.
> > > - Regarding the regeneration, while in the fishery scenario it makes sense to double the amount, since it is related to the amount of fish available that can reproduce, for the trash scenario, it doesn't make sense that the amount produced in the next month depends on the leftovers of the previous month. However, we have tested this, as the first version, and what happens is that the agents take the entirety of the trash in the first month, so the duplication doesn't occur. We agree that the solution would also pass to have a fishery scenario with linear dynamics for a closure comparison; however, since the amounts used for the fishery scenario in Piati et al were small on an exponential scale, the regeneration can be seen as close to linear.
> > > - This was empirical, since if not pointed out that the trash would make the house uninhabitable, the models wouldn't understand that the environment would collapse, while in the fishery scenario, they understood that the amount of fish would become 0: basically, the models were assuming that they could have infinite trash at home.
> > > - Regarding the reviewer’s note on the wording shift (“start” vs. “beginning”), we confirm this was an unintentional typo introduced during scenario adaptation. Thank you for catching it, but we don't see a major implication of that typo on the study.
> > >
> > > - The agents respond in sequence, and as in Piatti et al., each speaking agent selects who speaks next. The verbosity and apparent repetition observed in some dialogues appear to be artifacts of the models themselves rather than the simulation structure. Despite system prompts explicitly instructing agents to avoid repeating ideas, we found that certain models tend to generate similar responses across turns. We hypothesize that this occurs because each agent’s prompt includes only a minor delta from the previous agent dialogue, leading the model to stay close to the original phrasing or structure. Backing this, we noticed that if Agent B speaks after Agent A and if Agent B's model is different from Agent A's model, we observed a more different answer, while if they have the same model, we basically see a subtle gradient shift from C to B to A.

---

### Decision · Action_Editor_w9on · 2025-08-11

**Recommendation:** Reject

**Additional Comments:**

There were some presentation problems in the original submission: there were simply too many models and extensions (cross-lingual evaluation) that the paper was hard to read and lesee focused. The authors have largely fixed in their revisions by removing some of the analyses that were less clear.

One major problem remains, which is summarized by a reviewer's comment in their official recommendation:

> The "loss aversion framing" extension (one of two extensions) modifies the original prompt from a gain-maximization to a loss-minimization task, and compares the resource-sharing behavior of agents in this new setting. However, the changes made to the original prompt (a) also change the utility function and setting dynamics, and (b) also change the task setting in a way that could reasonably change the collaboration behavior. Therefore, I think the evidence is insufficient to conclude that the change in the agents' collaboration behavior is due to the loss framing as the authors claim. The fact that the authors only consider a single prompt setting makes the strength and generalizability of this finding additionally unclear.

Another reviewer mentioned this in their review as the first weakness:

> In the negative framing scenario, it is not clear if the change in behavior was due to the polarity of the framing other factors since multiple aspects of the scenario change.

The authors did not respond to this point, which the reviewer noted in their official recommendation. This is unfortunate, considering it was a major point also brought up by a different reviewer.

As a way to justify the evidence to support their claims, the authors did reply to discuss this point, including a hypothesis to explain the results. However, the reviewer remain unconvinced.

I recommend the authors provide stronger evidence (and more results) to further substantiate their hypothesis, and to elaborate on this specific point that was pointed out by 2/3 reviewers.

As the reviewers were happy with the rest of the paper, including some quite positive comments overall, I believe this paper would be ready to accept once this one problem is rectified. As such, I encourage the authors to resubmit the paper with more convincing evidence to support the claims about the loss aversion.

**Audience:**

Yes

**Audience Explanation:**

Yes, this is a clear audience fit.

**Claims And Evidence:**

No

**Claims Explanation:**

All three reviewers agree that the reproducibility part of the paper was well-done and is convincing.

The paper in its current form has one major problem, which is that the claims made about the loss aversion extension are on sufficiently substantiated by evidence.

More detail in the "Additional Comments" section.

**Resubmission Of Major Revision:**

The authors may consider submitting a major revision at a later time.